# Sensors and Measurements for Unmanned Systems: An Overview

**DOI:** 10.3390/s21041518

**Published:** 2021-02-22

**Authors:** Eulalia Balestrieri, Pasquale Daponte, Luca De Vito, Francesco Lamonaca

**Affiliations:** 1Department of Engineering, University of Sannio, 82100 Benevento, Italy; daponte@unisannio.it (P.D.); devito@unisannio.it (L.D.V.); 2Department of Computer Science, Modeling, Electronics and Systems (DIMES), University of Calabria, 87036 Rende, CS, Italy; f.lamonaca@dimes.unical.it

**Keywords:** unmanned systems, UAV, UGV, USV, UUV, sensors, payload, challenges

## Abstract

The advance of technology has enabled the development of unmanned systems/vehicles used in the air, on the ground or on/in the water. The application range for these systems is continuously increasing, and unmanned platforms continue to be the subject of numerous studies and research contributions. This paper deals with the role of sensors and measurements in ensuring that unmanned systems work properly, meet the requirements of the target application, provide and increase their navigation capabilities, and suitably monitor and gain information on several physical quantities in the environment around them. Unmanned system types and the critical environmental factors affecting their performance are discussed. The measurements that these kinds of vehicles can carry out are presented and discussed, while also describing the most frequently used on-board sensor technologies, as well as their advantages and limitations. The paper provides some examples of sensor specifications related to some current applications, as well as describing the recent research contributions in the field.

## 1. Introduction

Thanks to technology advancements, several systems or vehicles not requiring the physical presence of human operators on-board, and with increasingly autonomous functions, have been developed. These systems try to overcome human limitations, to satisfy different application demands more efficiently, and to bring out new tasks and applications. Unmanned systems, compared to manned systems, in fact, provide advantages such as improved mission safety and reduced operational costs. They enable long-range operations beyond the detection ranges of human observers, as well as providing access to physically stressful and hostile environments in which humans cannot operate.

Unmanned vehicles can operate by embedding several kinds of sensors, making them capable of being fully fledged mobile measurement platforms. They can operate in different environments and conditions, with different levels of autonomy, singularly or in cooperation with other unmanned vehicles of the same or different type.

Measurements are critical both for unmanned vehicle motion and to carry out the mission for which it is called to work. The sensors used to collect the data required by unmanned vehicles to operate can be very different in terms of their technology and performance, with different measurement accuracy capabilities needing to be taken into account. Increasing the autonomy level of unmanned vehicle functioning increases the importance and the number of the required measurements, too. The application range for these systems is continuously increasing in number and complexity requiring new measurement capabilities and methods.

This paper presents an overview of the main unmanned system classes, highlighting the role and the relevance of the measurements, and also provides some examples of recent research contributions in this field. First, unmanned systems are defined, and a classification based on their environment of operation—air, ground or water—is discussed. Then, the role of measurement for each class of unmanned system is presented, also taking into account the different sensor technologies adopted. Next, measurements for unmanned system cooperation are briefly presented. Finally, conclusions are drawn.

## 2. Unmanned Systems

An Unmanned System (US) or Vehicle (UV) can be defined as an “electro-mechanical system, with no human operator aboard, that is able to exert its power to perform designed missions” [1].

UVs can be remote controlled (by a remote pilot) or can navigate autonomously based on pre-programmed plans or more complex dynamic automation systems [2]. They include vehicles moving in the air (Unmanned Aerial Vehicle or System—UAV, UAS, commonly known as “drone”), on the ground (Unmanned Ground Vehicle—UGV), at the sea surface (Unmanned Surface Vehicles—USV) or in the water column (Unmanned Underwater Vehicles—UUV), briefly described in the following subsections.

### 2.1. Unmanned Aerial Vehicles (UAVs)

UAVs, also called drones, are unmanned systems navigating in the air, able to survey wide areas, and are able to reach human-hostile environments, too. They can be remotely piloted or autonomously controlled [3,4].

There exist different types of UAVs, depending on the specific purpose the UAV is designed for. UAVs can differ in size from the order of centimeters to tens of meters, in weight from tens of grams to thousands of kilograms, in operational altitude from tens of meters to thirty kilometers, and in range from 100 m to 1000 km [5].

Rotor wing UAVs are a kind of aerial vehicle possessing enormous diffusion and that is attracting growing interest among researchers. They have vertical take-off and landing capabilities and are often developed in the form of quadcopters, thanks to their small size, easy control and high maneuverability.

UAV-related technology is continuously and rapidly evolving, and the number of applications for UAVs is growing exponentially, and includes real-time monitoring, providing wireless coverage, remote sensing, search and rescue, package delivery, security and surveillance, precision agriculture, and civil infrastructure inspection, as shown in Figure 1 [3,4].

The environment in which UAVs operate can strongly affect their mission results. Extreme wind, rain and storms can cause the UAV to deviate from its predetermined path or, especially in the case of small UAVs, not allow the vehicle to operate and take measurements. Problems related to weather conditions are further exacerbated in the case of natural or man-made disasters, such as, for example tsunamis, hurricanes or terrorist attacks [6]. Another problem related to the UAV operating environment arises from the possible presence of obstacles along its path, and this concerns both outdoor and indoor environments. Moreover, obstacles can be stationary or moving, making avoiding them more complex.

The altitude reached by UAVs is another important parameter that can be influenced by the environment conditions. If the elevation of the area of interest changes rapidly and significantly, for example in the case of steep terrain, the UAV must be able to follow these changes quickly, adapting and reaching the required altitude. Moreover, in the case of high-altitude applications, the UAV must be able to adapt itself to the changes in atmospheric density and temperature, preserving its aerodynamic performance.

UAV payload can affect vehicle navigation and measurement abilities, as well as the mission endurance and covered area, which are important requirements for applications when the UAV needs to operate for extended periods of time over large regions of interest [6].

The limited payload weight, altitude, and covered distance, as well as the influences of weather condition and handling obstacles, represent some weaknesses of UAVs (Figure 1) that research is attempting to address. The widespread popularity of UAVs is not replicated in other classes of UVs.

### 2.2. Unmanned Ground Vehicles (UGVs)

UGVs are unmanned systems operating on the ground. They are used for many applications (Figure 1), including space exploration, environment sensing, and search and rescue, and can have many different configurations, usually defined by the task they must perform, as well as the environment in which they must operate. UGVs have been developed in different sizes (varying from 500 g to 25,000 kg [7]) and configurations, typically linked to the mission they were designed to carry out [8].

UGVs are generally equipped with a controller and on-board sensors to observe the environment and make decisions autonomously or send the information remotely to a human operator [2,8]. Since UGVs’ on-board sensors cannot see what is beyond obstacles around them, these kinds of unmanned vehicle can be impeded by their reliance on line-of-sight sensing (Figure 1) [9].

Additionally, for this kind of vehicle, the operating environment is a source of some challenges. Dust, smoke and rain can strongly influence the UGV mission results, limiting the operational speed and the capability of avoiding possible obstacles, such as, for example, other vehicles, which in this case can be either static or moving. UGVs must be able to adjust their trajectory and speed in a suitable time to avoid collisions. The higher the UGV speed, the further the distance at which the obstacles have to be detected, and the shorter the time available in which to change trajectory or stop the vehicle, also taking into account that UGV braking distance to full stop increases with speed.

The type of terrain on which the vehicle moves is another environmental factor capable of influencing its mission results and operations. The traversed ground can be of highway, urban, country, or off-road types. Urban environments are in general more complex and dynamic, due to the presence of many cross-roads, vehicles and pedestrians, moving at different speeds and in different directions. Off-road conditions can be less complex when the terrain is firm and regular, as in the case, for example, of a desert, but can also be more complex when the terrain conditions are irregular and unstable, as in the case of forests [10].

Gravel, pavement, soil of varying firmness, mud, snow, ice, water, and vegetation of various heights and thicknesses are other challenging kinds of terrain that UGVs have to traverse, while also paying attention to possible bodies of water and mud that the vehicle must be able to avoid in a timely fashion in order not to risk getting stuck.

Environmental perception and UGV level of autonomy are the main characteristics being focused on with respect to current and future technological developments for this kind of unmanned vehicle.

### 2.3. Unmanned Surface Vehicles (USVs)

USVs are vessels operating on the surface of water, and can be remotely operated or autonomous. USVs can be realized in many different forms, depending on the specific application [4]. They can vary in weight from tens of kg to thousands of kg, with speeds that can vary from 1 m/s to about 20 m/s [11].

USVs can operate in conditions that are dangerous and risky for human safety. Moreover, this kind of vehicle is compact, and has low maintenance costs, too.

Although USVs were initially used in typically naval applications, such as, for example, surveillance and reconnaissance, nowadays they are also widely used in civil applications, such as, for example, environmental monitoring and assessment, as can be seen in Figure 1.

Autonomous shipping, search and rescue, offshore surveying in the oil and gas industries, seabed mapping, and inspection of structures above and below water are other examples of USV applications [4]. USVs need to operate in the presence of waves, currents and winds (Figure 1).

This kind of vehicle operates in contact with two environments, air and water, that have completely different physical characteristics. It is necessary for USVs to have sufficient environmental perception to detect and avoid obstacles that may be above or below the water, to estimate their motion and distance, and to perceive, in the case of coastal areas, the boundary between water and land.

Adverse weather and water conditions such as rain and extreme wind or rough and choppy water can strongly affect USVs’ mission results and operation.

Endurance for long-term missions and operation under extreme weather conditions, as well as above and underwater obstacle avoidance are the technological challenges that form the main subjects of research.

### 2.4. Unmanned Underwater Vehicles (UUVs)

UUVs operate under the surface of the water with minimal or no human operator intervention. UUVs can be of different types, varying in shape and size, depth ratings, payload, navigational capabilities, and control. UUVs can differ in length from little more than one meter to tens of meters, can operate at different depths varying from 200 to 6000 m, and at different speeds from about 0.5 to 4 m/s, with a weight of up to thousands of kg [12,13].

These vehicles can be Remotely Operated Vehicles (ROVs), controlled by a remote operator, or Autonomous Underwater Vehicles (AUVs), operating independently from direct human input [4]. AUVs are the most complex, having to rely on autonomous functions in a difficult environment such as the aquatic one.

UUV applications include persistent surveillance, anti-submarine warfare, underwater construction and infrastructure maintenance support, oceanography, hydrography and mine countermeasures, as shown in Figure 1 [4].

UUVs must operate in a harsh environment under high ocean current and heavy hydraulic pressure (Figure 1). The amount of light available underwater is often poor, due to the water particulates scattering light as it enters the ocean, or to turbidity as, for example, in the case of shallow coastal water. UUV navigation and maneuverability can also be strongly affected by ocean currents and water density. In some extreme cases, sudden changes in density can even impede the movement of UUVs through the water. Vehicle stability can also be compromised by the effect of the movement of water caused by wind or density variations in deeper waters [14]. UUV must also pay attention to the detection and avoidance of static or moving obstacles, as in the case of the other kinds of UVs.

Moreover, since maintaining a constant connection with remote ground operators can be highly difficult in deep waters, precise navigation capabilities represent another critical issue [13,14].

## 3. Measurements for Unmanned Systems

Measurements are essential for UVs. Their navigation capabilities and monitoring ability with respect to several physical quantities in the environment around them are strictly dependent on sensors and measurement systems, as well as on data processing algorithms.

The development of autonomous driving is also closely connected to the capacity for interpreting and analyzing information coming from the measurements provided by sensors or combinations of sensors of very different types (daylight and night vision camera, LiDAR, millimeter wave/ultrasonic radar, etc.), in order to take advantage of their different optimal working ranges and to collect information related to different dimensions of their surroundings [15].

UV performance strongly depends on how well they can perceive the environment in which they move, especially if UVs are asked to move independently without being able to rely on the intervention of a human operator.

Obstacle avoidance capabilities are important for all the UVs, regardless of whether they operate in the air, on land or on/in water. Sensor fault detection is another important topic for all kinds of UVs to ensure their safety and reliability.

Different or equal types of UVs can also cooperate to improve mission performance, while at the same time overcoming the limitations of specific or single UVs. In the following, the measurement issues for UAVs, UGVs, USVs and UUVs are briefly presented, along with some examples of sensors that can be used for UV navigation and missions.

It is important to highlight that the complexity of environmental conditions and the specific design constraints and requirements makes the selection of UV measuring equipment a very difficult task.

UVs can be of very different size, weight and cost, aimed at carrying out specific missions that can require a different numbers, types and combinations of sensors. These sensors can be mounted in different ways, and their measured information managed by specific, different and custom data processing algorithms.

Therefore, finding the optimal sensor specifications that can be considered valid for the multitude of tasks, applications and types of UV represents an unsolvable problem. Individual sensor specifications and characteristics certainly affect the performance of the whole UV, but these are also determined by other factors, including their operational conditions and environment. Therefore, testing procedures are required to assess the suitability of the resulting UV for the design specifications and application requirements. Moreover, sensor technology is the subject of intense research activities and continuous advancements.

For these reasons, the following sections aim to give some general indications, including advantages and limitations, regarding the measurements and sensors typically carried and used for UAVs, UGVs, USVs and UUV, providing some examples of sensor specifications for specific applications, as well as some UV designs found in the literature.

### 3.1. Measurements for UAVs

UAVs’ environment is the air; they require location and attitude sensing, ground and air speed, angle of attack, barometric pressure, and sometimes need to sense location data or communicate this with other aircrafts. The flight position and orientation of UAVs are determined by using accelerometers, combined with tilt sensors and gyroscopes. The position and orientation information is then provided to the flight control system to maintain level flight [16,17]. Flight paths and directions are managed by using inertial measurement units (IMUs), combined with Global Positioning System (GPS)/Global Navigation Satellite System (GNSS) [16,17].

The precision of navigation-related measurements is a highly critical issue for UAVs, especially for complex flight phases, such as landing. Landing demands great precision, since altitude inaccuracies can lead to inaccuracy on the landing spot, while meter-level navigation inaccuracies can also lead to collisions with surrounding objects [18]. Moreover, IMU measurements can be corrupted by noise and bias, resulting in unreliable pose estimates for long-term navigation, leading the UAV position estimation to diverge and drift over time [19].

Other observations can be drawn with respect to the GPS/GNSS signal, since it can be unreliable in dense forests or urban canyons, and in indoor environments, the signal can be too weak, or almost lost, due to buildings, walls or several other potential sources of interference or jamming [19].

The problem of the precise localization of UAVs in indoor environments is gaining increasing interest from researchers.

Many indoor UAV solutions use an optical camera, usually together with other technology, such as, for example, UltraSonic (US) technology (Table 1) [20,21].

By means of a Time-of-Flight (ToF) camera, in [21], an initial estimation of UAV altitude was carried out, and then the horizontal vehicle positioning was computed through a 2D multilateration procedure by using an encoded Ultrasonic Local Positioning System (U-LPS). The proposed solution, as shown in Table 1, can achieve a difference between the estimated position and the actual one on the order of centimeters.

Ultra Wide Band (UWB) is another technology widely used in 3D indoor positioning, also, in this case, together with other technologies capable of providing centimeter accuracy.

In Table 1, the results achieved by a system based on time-difference of arrival (TDoA) UWB, using UWB receiver nodes and a tag located in the top of the UAV, integrating sensors such as optical-flow IMU sensors for velocity and yaw feedback, are shown. This solution also presents good accuracy, but requires high costs [22].

Due to the fact that a Wireless Local Area Network (WLAN) is commonly available in many indoor environments, it can be used to estimate UAV location without the need to meet high costs. An algorithm for mini-UAV indoor localization based on distance measurements between the UAV and existing infrastructure consisting of WiFi Access Points (APs) was proposed in [23]. A site survey is not required, since it is locally performed by the UAV, and the location can be obtained in real time [23]. This method shows good results in non-complex environments (Table 1).

Another technology that can be used is based on visual odometry; specifically, in [24], the proposed indoor UAV location solution relied on an ARM-based stereo vision pre-processing system, used as an on-board sensor capable of continuously estimating the UAV pose with six degrees of freedom. The solution has the advantages of small size and weight; however, the UAV location performance is dependent on light conditions [20].

Visual odometry is also used in [25], but this time in combination with UWB sensors as an indoor UAV location solution. In particular, UWB sensor information is merged by means of a Monte Carlo Localization algorithm with visual odometry on an RGB-D camera to estimate the UAV pose [20]. Although the proposed solution presents good accuracy, it has the drawback of high costs [20].

Infrared beacons, computer vision and IMU sensor data fusion have been proposed to address the problem of autonomous UAV landing, which requires precise vehicle localization under real-time constraints [26]. Depending on the landing point distance, the accuracy of this solution varies from ten centimeters to a few meters [20].

UAV performance and covered distance are dependent on the power consumption; therefore, sensors can be used to monitor and optimize it [16], as well as current sensing capability, for the determination of battery capacity during the charging and discharging process, and providing an alert for system faults occurring during the flight [17]. Measurements are also important for guaranteeing UAV safety and reliability for fault detection.

By using measurements from commonly implemented UAV sensors, such as gyros, accelerometers, GPS and wind vanes, it is possible, for example, to detect additive step faults in the measurements of the airspeed signal [27].

Measurements also play an important role in UAV testing [28]. Testing involves all the stages required for UAV development and use. It can be carried out both in a controlled environment (laboratory), and in the field, where the environment cannot be entirely controlled, and where the testing phase will be more complex and challenging [28,29,30,31,32,33].

When flying outdoors, for example, UAVs are vulnerable to weather conditions such as rain or windy weather. Therefore, the development of wind-resistant test equipment for UAVs is a subject of interest, too [34].

Thanks to the ability to collect a large amount of information, covering a wide field of view, great interest has been directed towards indoor flight testbeds using the vision method.

Consequently, advanced approaches and algorithms aimed at developing UAV indoor flight testbeds have been proposed [35,36,37,38,39].

Another important observation concerns the fact that, nowadays, UAVs are used as mobile measurement platforms for a wide range of different applications; however, measurement uncertainty issues have not yet been adequately and thoroughly dealt with [40,41].

Some examples of UAV sensors, including those used in agriculture for the monitoring and management of crops, in archaeology to provide a visual view of the site, document archaeological excavations, and conduct exploratory surveys and aerial reconnaissance, and in general applications to ensure UAV sensing and avoidance capabilities, are shown in Table 2. Red Green Blue (RGB) digital cameras provide high-spatial-resolution radiation values in the red, green and blue spectral bands [42]. The spatial resolution of the RGB sensor determines the quality of the acquired images. By using this kind of sensor in agriculture, measurements related to plant coverage, plant height, and color indexes can be extracted by processing the aerial images obtained with the camera [42]. Spectral sensors can be categorized as either multispectral or hyperspectral sensors, depending on the number of spectrum bands and the width of each spectrum band, and they measure the radiance reflected, emitted, and transmitted from the target to acquire information [42]. This kind of sensor can detect wavelengths in the visible spectrum and in the invisible near infrared spectrum [42], and are used in agriculture applications to provide measurements related to the physiological status of the plant and vegetation index.

Thermal sensors detect the electromagnetic energy in the infrared (IR) wavelength range emitted by the target, converting it into an image. They are used in agriculture due to their ability to provide measurements related to plant surface temperature and crop water stress index [42].

As it can be seen in the table, RGB cameras can provide a better spatial resolution than other kinds of sensors; however, in the agriculture field, multispectral cameras have great advantages with respect to RGB ones in terms of extrapolated information, such as, for example, the ability to detect the invisible physiological status of the plant [42].

Hyperspectral cameras are not commonly used in agriculture applications, since they require integration with a greater number of other devices, including a battery, a frame grabber, and a data storage device, in order to operate suitably on UAV platforms, as well as also being heavy and large [42]. However, hyperspectral sensors are increasingly the subject of miniaturization, which will increase the number of applications that can benefit from their use [43]. The fluctuating environmental conditions in the air and/or the presence of different objects emitting or reflecting thermal infrared radiation can degrade the reliability of the thermal camera measurements, requiring periodic calibration [42].

Light detection and ranging (LiDAR) sensors measure the distance to one target point by illuminating it and analyzing the reflected light. This kind of sensor is capable of providing a wide field-of-view (FOV), that is, the angle covered by the sensor, and a high accuracy. However, the size and weight can present a problem in terms of UAV payload requirements.

LiDAR sensors are used in archaeology applications for providing measurements related to the effects that archaeological remains buried close to the surface have on the topography of a landscape [43]. They can provide detailed digital terrain and surface models covering vast landscapes [43].

Spectral sensors provide, in the archaeological field, measurements allowing the detection of landscape matrix contrast, which is useful for providing interpretations of archaeological significance [43].

Archaeology is also able to gather useful information through the use of thermal sensors, since it is possible to detect distinct variations between the features and the soil matrix in which they are embedded on the basis of their measurements [43].

Especially for UAVs flying at lower altitudes, static and/or dynamic obstacles can represent a critical problem that also puts the integrity of the vehicle at risk. These considerations are valid for all kinds of applications using UAVs. Some sensors that can help UAVs to acquire information about the presence of obstacles and the surrounding environment are reported, along with their detection range, in Table 2.

Radars detect obstacles by continuously emitting electromagnetic waves travelling at the speed of light. The presence of an obstacle is determined when the emitted waves are reflected back towards it. The obstacle distance is determined on the basis of the measured time difference between the emitted wave and the reflected wave.

Radar has a wide sensing range, and it can scan a target area quickly [44]. LiDAR has a smaller detection range than radar, but it has the advantage of determining not only the obstacle distance, but also the information regarding the range of the obstacle [44].

Electro-optic sensors are capable of providing the relative position of the obstacle by determining the elevation and azimuth by means of a camera, but do not provide distance or speed information.

Differently from radar and LiDAR, electro-optic sensor performance is strongly affected by weather and cloudy environments [44].

### 3.2. Measurements for UGVs

Environment perception is a fundamental task for UGVs, and requires the extraction of information from the measurements gathered from various on-board sensors. Typically, there are, in fact, many more obstacles on the ground than there are in the air or in the water. Moreover, UGVs can be required to operate under challenging conditions resulting from dust, rain or smoke.

For the state estimation system to monitor UGV platform status in real time, it is essential to gather measurements from sensors, too [45,46,47,48,49].

UGV sensors can be divided into exteroceptive and proprioceptive sensors [45,49]. Exteroceptive sensors are required for external environmental perception systems, and include LiDAR, radar, and ultrasonic sensors, as well as monocular, stereo, omnidirectional, infrared and event cameras. In Table 3, the main characteristics of these sensors are briefly shown.

LiDAR can be used to detect both the position and speed of a target by emitting a laser beam, and possesses a long detection distance range, a wide field of view, and high data collection accuracy. Moreover, since LiDAR is not affected by lighting conditions, it can operate during both night and day.

On the other hand, this sensor performance degrades in case of rainy, snowy, foggy conditions and sandstorm. Dark and specular objects cannot be rightly detected by LiDAR, due to their absorption or reflection of most laser beam radiation [45].

Radar sensors, unlike LiDAR, emit radio waves for detecting the position and speed of a target. Although radar is not affected by weather conditions like LiDAR, compared to the latter, they have a lower resolution and accuracy.

Ultrasonic sensors carry out target detection by emitting sound waves. This approach has low cost and size, but also low accuracy, and its narrow field of view can cause blind spots during target detection operations.

Monocular cameras convert an optical signal into an electrical signal by storing information by means of pixels; they can obtain high-resolution images with color and texture information from the surrounding environment. However, monocular cameras are strongly affected by weather and illumination conditions, implying a high calculation cost when obtaining high-resolution images.

Stereo cameras, in contrast to monocular cameras, are equipped with additional lenses at symmetrical positions, or can be capable of calculating the time of flight for generating depth information. They can extract color and motion information from the environment, but they are influenced by the weather and illumination conditions, as well as having a narrow field of view.

Omnidirectional cameras can collect a ring-shaped panoramic image centered on the camera. They have a large field of view, but also a high computational cost, being affected, in the same way as the previous kind of camera, by illumination and weather conditions.

Infrared cameras provide information regarding the environment on the basis of the infrared radiation emitted by objects. They have good performance at night, but they cannot provide color or texture information, and their accuracy is low.

Event cameras measure changes in brightness in each individual pixel of an image at the microsecond level. They are affected by weather and illumination conditions, but reduce the time required for information transmission and processing while maintaining a high dynamic measurement range. The output data resolution is low [45].

The highest ranges, as can be seen in Table 3, are achieved by LiDAR and radar sensors, and the smallest by ultrasonic sensors, while in the case of monocular, omnidirectional, and event cameras, the range values are dependent on the operational environment.

Proprioceptive sensors are equipped for UGV state estimation and include GNSS and IMU [45]. Sensor fusion, object detection at both very short and very long distances, as well as the possibility of providing information on surroundings from sensors at high speed are important topics of focus in UGV technology [8,45,50,51,52].

Terrain represents the biggest issue for UGVs, since it can be unpredictable, unstructured and untraversable, potentially causing the vehicle to become stuck, making it useless [8].

This problem also shows up in urban environments, since they include a wider variety of terrain classes and often mixtures of materials [49].

For these reasons, terrain traversability analysis methods have been proposed, using different sensor measurements, also considering the possibility of providing a system capable of sensing and measuring terrain using vehicle performance measurements [8].

In Table 4, some examples of sensors used to allow the UGV to perceive the surrounding environmental conditions in complex outdoor applications and on complex surfaces are reported.

Radar can detect a range up to 40 m, providing the best horizontal FOV.

Visual cameras can acquire images with a higher resolution than the infrared camera that provides the lowest FOV values [53].

Many challenging applications, such as, for example, surveillance, rescue, and planet exploration, require the autonomous navigation of the UGV in a non-planar environment, so it is necessary that the vehicle be capable of collecting information regarding the effects of its interactions with rough terrain surfaces; that is, terrain traversability.

Traversability information can typically be gathered using visual or LiDAR sensors. However, visual sensors require long computation times to process the visual data, and their performances are influenced by the intensity of sunlight [54].

In Table 4, some specifications of LiDAR sensor used for traversability assessment are shown, including a distance accuracy less than 2 cm and a range depending on terrain type, but that can reach a maximum value of 120 m (for cars and foliage) and a minimum of 50 m (for pavement) [54].

Due to their good performance for traversability assessment, LiDAR finds application in the agricultural field for providing reliable navigation inside orchard rows, as it is capable of detecting rows of trees, obstacles, branches, ditches, and other non-transitable areas [55].

An example of a LiDAR sensor used for orchard management is shown in Table 4.

In agricultural applications, visual sensors, such as, for example, RGB cameras, can be used for the automatic registration of fruit locations [56] and for fruit detection, which can also be achieved using thermal camera [57].

### 3.3. Measurements for USVs

Measurements provided by sensors are essential for USVs and have a strong impact on their performance. The marine environment can be affected by disturbances such as, for example, wind, waves, currents, sea fog, and water reflection; therefore, environmental perception is crucial for USVs [58].

Passive perception methods can be implemented by using visual/infrared sensors measurements and active perception methods by using LiDAR, radar, and sonar [58].

USV state estimation is equally important, and can be provided by conventional GPS-IMU-based approaches [58]. However, in practical applications, effects arising from environmental noises, accumulated inherent drift errors, time-varying uncertainties arising from the varying environment, payload, and operating conditions must be taken into account. All these effects, in fact, together with sensor faults due to, for example, salt spray and moisture sensor damages, can lead to GPS-IMU systems not being suitably precise, requiring some additional correction actions to improve navigation performance [58].

For these reasons, other sensor-based approaches are usually adopted, such as, for example, active ranging sensors (LiDAR, radar and sonar), especially in cases of the loss or jamming of GPS signals, or vision-based approaches [58].

Particular attention has been devoted to obstacle detection, representing one of the most important and challenging requirements for achieving autonomous USV operation. Significant variations in lighting and constant tilting of the vehicle, the possibility of having a large part of an obstacle submerged, rapidly changing water surface, reflections, and the absence of texture in clear water make the operation of USV sensors to correctly avoid obstacles very complex [59]. Moreover, the environmental information provided by sensors needs to be continuously updated, increasing the calculation load [60].

Inertial sensor measurements can be used to improve USV vision-based obstacle detection, while multiple sensor measurements can be used to implement a dynamic obstacle model. Information such as the shape, speed and position of obstacles or for reducing the tracking errors in obstacle detection can be provided to improve reliability of the USV’s surrounding environment analysis [59,60,61].

Therefore, to improve USV performance, while also considering the harsh restrictions and requirements imposed by the marine environment, multiple heterogeneous sensors are usually employed. In this way, it is possible to make the best use of the different sensor characteristics.

In Table 5, the main characteristics of some on-board USV sensors are summarized with respect to their advantages and limitations.

High depth resolution and accuracy are provided by radars, LiDARs and sonars, but their performance can be degraded due to fast turning maneuvers, USV and environmental motion, and the noise from near the surface, respectively [58].

Low weight and small size, as well as low power consumption, are important advantages for USVs, and can be achieved through the implementation of IMU, GPS, visual and infrared sensors. However, IMU and GPS are sensitive to the magnetic environment, while visual sensors exhibit low depth resolution and accuracy, and the performance of infrared sensors can be degraded due to interference and depending on the distance [58].

As stated above, USVs must be capable of promptly detecting obstacles, collecting information from various sensors in order to reliably carry out avoidance actions for the safe and successful execution of the mission. In Table 6, some examples of sensors used to realize USV anti-collision systems are reported, along with their main specifications. Radars can detect obstacles, also determining their position, direction and speed [62]. This kind of sensor can detect at long ranges, and provide high depth resolution, accuracy and speed estimates, but the quality of the data can be affected by fast turning maneuvers, high waves, and water reflectivity. Moreover, in the case of small and dynamic obstacles, radars can provide limited detection capabilities [62].

LiDARs can provide the spatial representation of the USV surface area and determine the movement speed and path of the surrounding obstacles. This kind of sensor exhibits good obstacle detection at close range, as well as high depth resolution and accuracy, but it is sensitive to environmental and vehicle motion [62].

Visual sensors, other than providing the obstacle localization, are capable of differentiating the detected obstacle types according to their shape, color and texture.

Visual sensors can provide high lateral and temporal resolution and are simple with a low weight, but with respect to radars and LiDARs, they have a lower depth resolution and accuracy. Moreover, light and weather conditions can strongly affect the performance of this kind of sensor. However, visual sensors can improve the reliability of anti-collision systems, supplementing LiDAR and radar measurements [62].

In Table 6, some examples of sensors that can be used for realizing low-cost USVs of small size are also shown. Although the LiDAR sensors have a lower range and accuracy with respect to the other LiDAR sensors described previously in the paper, they have a very low weight and a small size.

The visual camera also has small dimensions, with relatively high resolution [63].

In Table 6, two examples of sensors used in archaeology applications are also shown.

The first is sonar sensors, which ware widely used for bathymetric measurements to acquire information on the topography of the seafloor [64].

The second is a visual camera with shockproof and waterproof protection, used to check the underwater environment and the presence of obstacles in real time.

Ultrasonic sensors are also used for obstacle detection, often together with other sensors such as visual cameras, to perceive, for example, scattered rocks, which are sometimes not cartographically marked [64], or thermal sensors to compensate for the effects of temperature variation on the accuracy of distance detection [64].

In Table 6, some examples of sensors used by USVs to detect oil on or near the surface of the water are shown. These sensors do not require direct contact with the oil in the water [65].

Thermal IR cameras can reveal the contrast between oil and water through the measurement of emissivity and temperature. Oil radiates less thermal energy than seawater at the same temperature, since it has a slightly lower emissivity. Therefore, in an IR image, under nighttime conditions, when the oil and the water are the same temperature, the oil appears cooler than water, depending on the oil–water emissivity difference. Under sunlight conditions, the oil appears warmer than the surrounding seawater instead in IR images [65].

However, low visibility conditions, such as, for example, rain and fog, can degrade the IR camera performance.

An example of radar, used for oil detection, with its main specifications is shown in Table 6.

Radars can detect oil on the water by looking at the way the oil absorbs localized wave movement at the surface [65]. In particular, a radar return, known as “sea clutter”, is generated by waves and suppressed in the presence of oil. A “dark” region of the radar return indicates the detection of oil, because in this case, the cross section is smaller than for normal surface waves [65].

The presence of a measurable “sea clutter”, which is ensured with wind speeds within the range from 1.5 m/s to 6 m/s, is required to allow the radar to discriminate many oil spills.

If the wind speed is too low, the “sea clutter” is too weak to be detected, while if the wind is above the required range, the too-large waves can disperse the radar signal, degrading the oil detection capability [65].

### 3.4. Measurements for UUVs

Due to the attenuating effects of water, UUVs, unlike USVs, do not rely on GPS signals when not resurfacing, instead relying on acoustics, sonar, cameras, Inertial Navigation Systems (INSs), or different sensor combinations to navigate [66,67].

UUVs’ target and trajectory tracking represents a very important requirement in order to allow the unmanned vehicle to carry out its mission [68,69].

Table 7 shows the capabilities, advantages and limitations of the main sensor typologies adopted for UUV operations.

Inertial navigation by means of data provided by sensors can determine the positioning and heading of the UUV; therefore, INSs are a payload commonly used aboard this kind of UV.

Due to the fact that high precision can be achieved by using INS only for short periods, INS data can be fused with those coming from other sensors, such as, for example, Doppler Velocity Logs, (DVLs), to provide more accurate measurements.

Long baseline (LBL), short baseline (SBL) and ultrashort baseline (USBL) acoustic sensors can be employed to allow the UUV to navigate by means of single or multiple transponders working at different frequencies.

The frequency of the involved signals affects the system accuracy; specifically, the higher the frequency, the higher the accuracy. As a consequence, ultrashort baseline sensors have the highest frequency and long baseline sensors the lowest. However, at the higher frequencies, the range turns out to be decreased.

DVL systems can measure UUV velocity and correct INS drifting errors, but only if the UUV operates in close proximity to the seafloor to establish a bottom lock [66].

UUV depth is important information for safeguarding the vehicle from operating at depths capable of damaging their functionality [66]. UUV depth measurements are provided by pressure sensors. In particular, the ambient pressure in the water column can be calculated through measurements provided by pressure sensors such as strain gauges, providing measurements accurate to within 0.1 percent [66], and quartz crystals, since their resonant frequencies are correlated with ocean pressures, capable of determining depth to an accuracy of 0.01 percent [66].

UUV orientation can be determined by compasses and magnetic, roll-and-pitch, and angular-rate sensors.

Magnetic sensors can be affected by systemic errors; roll-and-pitch sensors can be accelerometers, pendulum tilt sensors, or fluid-level sensors capable of determining UUV orientation with reference to gravitational forces. When surges in acceleration occur, low-cost sensors can be affected by performance degradation. With respect to information related to the angular rate, this can be obtained using gyroscopes [66].

Light and optical sensors are affected by poor accuracy due to the attenuation of light in the water column [66].

Optical sensing underwater, in fact, is very complex, since the optical wavelengths are scattered, diffused, and distorted by both completely clear water and the cloudy, turbid water of the ocean. Moreover, the effects of the low-light underwater environment, and the multiple layers of light refraction taking place among the sensor, the glass covering the sensor, and the water between the sensor face and the targeted object have to be taken into account [66].

Sensor calibration, along with sufficient processing capability for error correction and the close proximity of the sensor to the targeted object, along with the recording of multiple target images in varying light conditions or from varying locations, have been proposed as solutions for overcoming the optical sensing challenges [66].

Sonar systems can identify specific landmarks in the environment for the UUV localization [70]. By means of sonar, it is also possible to calculate the distance from the target and the relative speed of the target compared with the sonar source.

Information provided by sonar includes target identification and tracking, as well as the 3D imaging of a target [66].

Since sound propagation in water varies with temperature and salinity, sonar requires correct calibration [70].

Accurate UUV location and motion estimation requires predicted target states to be obtained by means of some known a priori knowledge regarding the target.

Measurements of target states provided by sensors can be used to modify the previous target states by using a certain algorithm [69].

A very recent proposal for achieving high-precision UUV state estimation is based on a Gated Recurrent Units (GRUs)-based deep neural network framework capable of predicting the UUV states based on previous measurements [69].

However, UUV sensors are also vulnerable to faults, which can lead to undesirable abortion of the mission, and even the loss of the vehicle [68].

A sensor active fault-tolerant control scheme was proposed in [68]. By means of this control scheme, it is possible to carry out UUV trajectory tracking in cases where the vehicle is subject to disturbances and several system uncertainties are present.

The proposed solution is based on an dynamic UUV model moving in the horizontal plane and a sensor fault model [68].

UUV operational endurance is a crucial parameter for long-range survey missions; therefore, there is great interest, on the one hand, in improving the vehicle’s energy storage and/or decreasing its consumption, such as, for example, by making use of advanced battery technologies (e.g., lithium-air and zinc-air batteries), in order to address the problem of high costs, which can be prohibitively expensive, or, on the other hand, in making use of very low power propulsion systems, which have disadvantages in terms of sensor payload capabilities, which are limited as a determination of their considerable operational applicability constraints [71].

When UUVs must manage a large volume of data, their communication can be a problem due to the limitations stemming from the low bandwidth of their acoustic communications while they are operating underwater.

For these reasons, a mothership must periodically recover and recharge the UUV, significantly limiting vehicle efficiency [71].

Automated UUV launch and recovery docking guidance systems can enable the vehicle to operate fully autonomously from an underwater dock, providing battery recharge facilities, as well as data download and upload, avoiding the necessity of recovery of the UUV by a ship [71].

It is essential for these systems to have some kind of docking guidance system assisting the UUV when performing suitable homing and docking operations in different environmental operating conditions.

In Table 8, some examples of sensors used in this kind of guidance system are reported, along with their acquisition range.

Acoustic sensors can provide acquisition ranges on the order of kilometers and high accuracy, as well as also being insensitive to oceanographic conditions.

However, the performance of this kind of sensor is influenced by acoustic refractions, sound reflection reverberation and ambient noise, the underwater physics of sound, which causes latency, and the dependence of range and direction measurement reliability on distance.

Optical sensors provide a limited acquisition range and are affected by signal attenuation due to the scattering and absorption of light, and interference from other light sources, but they provide high lateral and temporal accuracy.

Electromagnetic sensors provide a relatively short acquisition range, but also a high accuracy, ensuring good performance for almost all oceanographic conditions [71].

As quoted above, localization and navigation UUV capabilities are essential for the suitable operation of the vehicle, especially in the case of collaborative missions, such as surveillance and intervention, implying the collaborative work of multiple UUVs.

In Table 8, some examples of sensors used for UUV localization and navigation are presented, along with their specifications in terms of accuracy and range.

Sonar sensors are used to detect obstacles, such as seabed changes, rocks, marine species or other vehicles.

They can provide a minimum accuracy on the order of several centimeters up to about one hundred centimeters, and a range of up to hundreds of meters from obstacles.

Acoustic ranging sensors provide information about UUV position; their accuracy can vary from a few centimeters up to tens of meters.

Optical sensors provide information regarding position and orientation relative to a target to the UUV. These kinds of sensor include optical cameras and optical light sensors.

Although, due to the poor light transmission through water, imaging systems are affected by limited range, different algorithms and techniques have been developed to improve optical sensor measurement capabilities [72].

In the Table, it can be seen that optical light sensors can provide an accuracy of up to 20 cm for UUV position and 10° for orientation, while the optical camera exhibits better values, providing an accuracy of up to one centimeter for position and 3° for orientation. The range for both kinds of optical sensor is the same [72].

UUVs are widely used for deep-sea topographical exploration.

In Table 8, two examples of sonars that make it possible to obtain relatively accurate seafloor topographical data, providing measurements of size, shape and height variation of underwater targets, are shown [73].

The MultiBeam Echosounder (MBE) uses acoustic beams with known directions to estimate the backscattering of the seabed.

The Side Scan Sonar (SSS) provides a map of seabed acoustical reflectivity by measuring the acoustic surface reflectance of the seabed [74].

### 3.5. Measurements for Unmanned System Cooperation

UAVs, UGVs, USVs and UUVs have their own characteristics, advantages and limitations. To overcome their constraints or limitations, to improve their performance, to add new capabilities and functionalities, and to extend their use in new types of applications, multiple UVs can be made to work together, taking advantage of the use of several UVs of the same type, or the collaboration of UVs of different types.

For example, multiple UAVs or USVs can cover larger areas more efficiently than a single UAV or USV can do (Figure 2).

UGVs and UAVs can cooperate so that, for example, the UGVs can provide UAVs with the location of targets on the ground when high accuracy is required (e.g., the location and detection of dirty bombs, or other devices with the potential to disperse radioactive material, especially in urban zones) [75,76]; on the other hand, the UAVs can enable the UGVs to negotiate obstacles [75].

Moreover, USVs can be used for tracking UUVs and providing them with accurate real-time navigation information. In this case, measurements also play a critical role; sensor measurements and failures, in fact, are capable of strongly affecting the UV cooperation results and capabilities.

In the case, for example, of UAV swarms, accurate navigation information is an essential requirement for guaranteeing the UAV swarm’s successful completion of the targeted task.

Although satellite navigation is the main method of UAV swarm positioning, in urban areas, satellite signals may not be available, or the cost of high-precision satellite navigation equipment can become prohibitive when higher positioning accuracy is needed to meet the mission requirements, such as, for example, in the case of dense clusters [77].

These are some of the reasons that have led research to focus on cooperative navigation methods for the purpose of improving UAV swarm positioning accuracy, by means, for example, of visual assistance or wireless ranging information assistance [77].

Great interest has also been directed towards the development of small drones or Micro Air Vehicle (MAV) swarms [78,79,80,81,82]. MAV, thanks to being small and lightweight, can be simply designed, provide high portability, and navigate easily in tight spaces, such as, for example, restricted indoor environments. However, their smaller size constitutes a heavy limitation of their capabilities in terms of equipped sensors, flight time and power use [78].

MAV swarms can overcome the limitations of singular behicles, but present some challenges that need to be addressed.

Measurements are essential for allowing MAVs to gather information about their on-board states, in terms, for example, of attitude, velocity, height and altitude. Moreover, MAVs belonging to the swarm need to determine information sourced from their on-board sensors in order to quickly identify and avoid obstacles that could be present in the environment. When working in swarms, MAVs need to solve the problem of relative localization [78,83,84,85,86,87,88,89].

Technologies and sensors for carrying out direct MAV-to-MAV relative location estimation are currently the subject of research, and are continuously being developed.

Some examples of technologies that can be used for MAV relative localization are presented in Table 9.

Relative localization can be achieved by means of the visual detection of nearby MAVs. This is a complex task due to the different angles, positions, speeds, and sizes of the MAVs to be detected, and any unfavorable lighting conditions.

One solution could be the use of visual aids affixed to the MAV, such as, for example, visual markers or colored balls [78,83,84,85]. However, the localization accuracy is still dependent on light conditions and visual line of sight, which also results in a heavy computational load.

The use of active markers, such as, for example, infrared or ultraviolet sensors, can improve the accuracy and decrease the effects of the light conditions, but visual line of sight is still needed, and using many active sensors can result in high levels of energy expense [78,86,87].

MAV relative position can be estimated by means of microphones to hear nearby vehicles. This solution allows operation in visually cluttered environments in an omni-directional way.

However, the sound emitted by the listening vehicle could obscure the sound of nearby MAVs, since they are similar. Therefore, the accuracy of this kind of relative localization solution is related to how similar the sounds of other MAVs are, as well as to the microphone setup.

When a microphone array is used, the accuracy of the localization is dependent on the length of the baseline between microphones [78,88].

An array of infrared sensors (emitting and receiving) can be used and placed around the MAV. In this way, omni-directionality can be achieved, while also obtaining an accurate and computationally simple solution; however, this requires visual line of sight, and can lead to high energy expense, resulting in a relatively heavy system [78,89].

The use of more than a single UV can also be useful for improving the navigation performance.

An example is discussed in [90] in which an original in-flight calibration approach is proposed to estimate magnetic biases caused by UAV on-board disturbances, by combining the navigation data acquired by two vehicles.

The “chief” UAV, whose magnetometers are calibrated, uses a visual camera to detect and track the second, “deputy”, UAV within a sequence of frames, thus obtaining relative line-of-sight (LOS) information in camera coordinates.

Moreover, the two UAVs must fly under nominal GNSS coverage to enable relative positioning. The fusion of visual and GNSS data allows the determination of the magnetic biases by means of a system of non-linear equations that can be solved using the Levenberg-Marquardt algorithm [90].

In Table 10, some examples of sensors that can be used to avoid possible collisions, not only with environmental obstacles, but also among all of the vehicles belonging to the swarm, as well as to provide information about UAV relative localization in the case of both outdoor and indoor environments are shown.

The examples of obstacle detection sensors for avoiding collisions include an infrared sensor with a range up to 150 m, and an ultrasonic sensor with a shorter range, up to about six meters, but with a resolution of 2.5 m [91].

In Table 10, an example of visual camera that can be used for the UAV relative localization is also reported, along with its main specifications [92].

## 4. Conclusions

Today, unmanned systems are used successfully in a variety of applications, both civil and military. Measurements are crucial for the functioning of these systems, especially if they have to operate autonomously in the air, on the ground, or on or in the water.

The paper presents some examples of the measurement issues and research contributions for UAVs, UGVs, USVs and UUVs, to highlight how the present and the future applications and performance of these vehicles are and will be closely linked and influenced by the implemented measurement systems and methods. Future challenges for UVs concern the possibility of increasing their level of autonomy as much as possible, until they are completely autonomous.

Much of the research is aimed at overcoming the typical limitations of each type of UV, while also looking at solutions based on systems formed by several UVs of the same type that work together or as heterogeneous systems formed by UVs of different types that collaborate by compensating for the limitations of each other.

Another interesting aspect is related to the continuous extension of the number and type of applications that can benefit from the use of these systems. Particular attention is also paid to the use of UVs in indoor and/or urban environments or where GPS signals cannot be reached.

The considerable diffusion and growing potential of UVs can even make these systems dangerous, especially in cases where they are used incorrectly or for malicious purposes. For these reasons, some research work is also devoted to developing measurements and sensors aimed at identifying and eventually neutralizing UVs.

The last, but not the least, aspect to be considered concerns the impact of UVs on the economic growth and technological development of each country. This has led to increasing demand for professionals to design, develop, integrate, test and correctly use UVs, leading to the need to introduce specific training courses in this field [93]. UV education is also important for developing countries, since they can harness UV technology for humanitarian and development goals.

Although different teaching courses focusing on UAVs have been developed and provided at different educational levels [94,95,96,97,98,99,100,101,102,103], the poor availability of educational resources concerning the specific field of measurement for and by UVs can be observed.

Such educational topics are very important in the case of UVs for a variety of reasons.

First, UAVs, UGVs, USVs and UUVs, due to their unmanned, need to measure the environmental conditions by means of sensors and data acquisition systems. In particular, they are required to measure their internal functioning parameters, both mechanical and electrical, to dynamically control their movement, position and speed.

Second, UVs are equipped with sensors aimed at measuring certain environmental quantities of interest for the targeted applications they are used for, working as real flexible and mobile measurement instruments.

Finally, the safe use of UAVs, UGVs, USVs and UUVs must be ensured, and this requirement implies the need for measurements for the purpose of carrying out compliance tests.

## Figures and Tables

**Figure 1 sensors-21-01518-f001:**
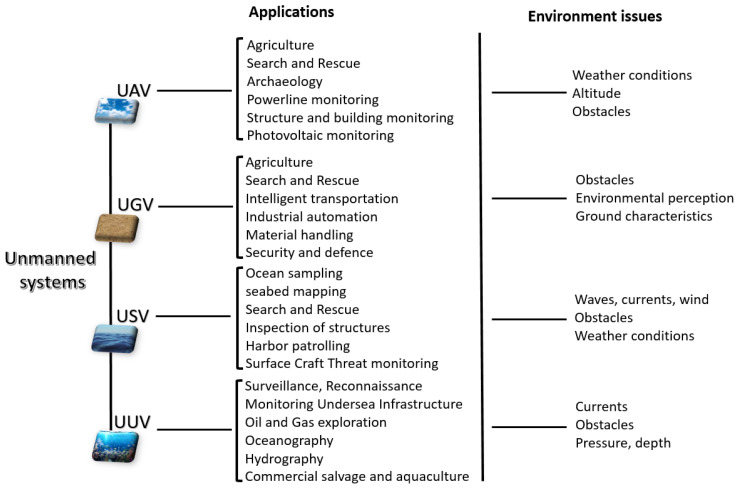
Unmanned system types and their critical environmental factors.

**Figure 2 sensors-21-01518-f002:**
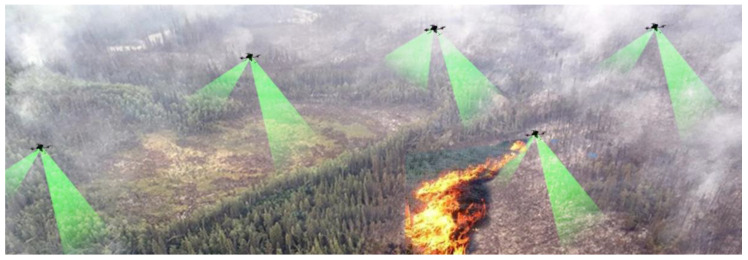
An example of UAVs collaborating for forest monitoring.

**Table 1 sensors-21-01518-t001:** Some indoor UAV location solution proposals [20].

Used Technology	Difference between the Estimated Position and the Actual One	Advantages and Drawbacks
Camera, US	cm (in static test positions)	Good precision/Doppler effect, acoustic noise
IMU, Optical, US, GNSS, UWB	10 cm	Good accuracy/High Cost
WiFi	1 m	Reuse infrastructure/Tradeoff between accuracy and complexity
Stereo Vision	10 cm	Size and weight/Light condition influence
UWB, RGBD Sensor	20 cm	Good accuracy/high cost
Infrared, IMU, computer vision	10 cm–4 m	Easy to deploy/sunlight interference

**Table 2 sensors-21-01518-t002:** Some examples of UAV sensor application specifications.

**UAV Application: Agriculture Monitoring and Management of Crops**		
**Functions**	**Sensors**	**Specifications**
Plant coverage, plant height, and color indices	RGB camera	Spatial resolution (1280 × 720); (1920 × 1080); (2048 × 1152); (3840 × 2160); (4000 × 3000); (4000 × 3000); (4056 × 2282); (4160 × 2340); (4608 × 3456); (5344 × 4016); (5472 × 3648); (5472 × 3648)
Vegetation indices; physiological status of the plant	Multispectral camera	Spatial resolution (1080 × 720); (1248 × 950); (1280 × 960); (2064 × 1544) Weight from 30 g to 420 g Frame rate from 1 fps to 30 fps
Plant surface temperature; Crop Water Stress Index	Thermal camera	Spatial resolution (336 × 256); (640 × 512); (1920 × 1080) Weight from 92 g to 370 g Spectral range from 7.5 μm to 14 μm Operating Temperature Range (°C) from −40 to 550
**UAV Application: Archaeology exploratory survey and aerial reconnaissance**		
**Functions**	**Sensors**	**Specifications**
Detailed digital terrain and surface models; penetrating vegetated landscapes	LiDAR	Range from 100 m to 340 m FOV (deg) (Vertical) 20, 30, 40 (Horizontal) 360 Accuracy from 1 to 3 cm Weight from 0.59 kg to 3.5 kg
Landscape matrix contrast detection	Multispectral camera	Resolution 1280 × 960; 1280 × 1024; 2048 × 1536; 2064 × 1544 Spectral range Blue, Red, Green, Near-infrared, Red-edge, long-wave infrared
Landscape matrix contrast detection	Hyperspectral camera	Resolution 640 × 640; 640 × 512;1024 × 1024; 2048 × 1088 Spectral range from 380 nm to 13,400 nm Weight from 0.45 kg to 2 kg
Detection of measurably distinct variations between the features and their soil matrix	Thermal camera	Resolution 160 × 120; 320 × 240; 320 × 256; 336 × 256; 382 × 288; 640 × 480; 640 × 512 Accuracy (±°C) from 1 to 5 Spectral range from 7 nm to 14 nm Weight from 39 g to 588 g
**UAV Application: General**		
**Functions**	**Sensors**	**Specifications**
Sensing and avoiding capabilities	Radar	Detection range 35 km
	LiDAR	Detection range 15 km
	Electro-optic sensor	Detection range 20 km

**Table 3 sensors-21-01518-t003:** Characteristics of exteroceptive sensors [45].

LiDAR	Radar	Utrasonic	Monocular Camera	Stereo Camera	Omni Direction Camera	Infrared Camera	Event Camera
High accuracy	Medium accuracy	Low accuracy	High accuracy	High accuracy	High accuracy	Low accuracy	Low accuracy
Range <200 m	Range <250 m	Range <5 m	Range operational environment dependent	Range <100 m	Range operational environment dependent	Range operational environment dependent	Range operational environment dependent
Affected by weather	__	__	Affected by weather and illumination	Affected by weather and illumination	Affected by weather and illumination	Affected by weather	Affected by weather and illumination
Large size	Small Size	Small size	Small size	Medium size	Small Size	Small size	Small size
High cost	Medium cost	Low cost	Low cost	Low Cost	Low Cost	Low cost	Low cost

**Table 4 sensors-21-01518-t004:** Some examples of UGV sensor application specifications.

**UGV Application: Challenging Environment Outdoor Applications**		
**Functions**	**Sensors**	**Specifications**
Reliable perception, obstacle detection	Radar	Maximum range 40 m Range resolution 0.2 m Horizontal FOV 360° Angular resolution ≈ 1.9°
	Visual camera	Image size 1340 × 1024 FOV (horiz.) 68.2°; (vert.) 53.8°
	Infrared camera	Image size 640 × 480 FOV (horiz.) 35.8°; (vert.) 27.1°
**UGV Application: Application operating on complex terrain surface**		
**Functions**	**Sensors**	**Specifications**
Traversability assessment	LiDAR	Distance accuracy < 2 cm Measurement range from 50 m to 120 m FOV (vert.) +2.0° to −24.8°; +10.7° to 30.7° vert. angular resolution 0.4°; 1.33° horiz. angular resolution 0.09°; 0.16°
**UGV Application: Agriculture**		
**Functions**	**Sensors**	**Specifications**
To detect rows of trees, obstacles, branches, other orchard features, ditches, tranches, other non-transitable areas	LiDAR	Range up to 100 m Accuracy ± 3 cm FOV (horiz.) 360°; (vert.) 30° vert. angular resolution 2° horiz. angular resolution 0.1°–0.4°
Automatic registration of fruits locations	RGB camera	Resolution 1600 × 1200 Frame rate 35.6 fps Color depth 12 bit
Fruit detection	Thermal camera	Resolution 320 × 240 Spectral range 7.5–13 μm Resolution 0.05 °C

**Table 5 sensors-21-01518-t005:** Characteristics of USV sensors [58].

Sensors	Advantages	Limitations
Radar	Long detecting range, nearly all-weather and broad-area imagery, high depth resolution and accuracy	Skewed data in fast turning maneuvers, limited small and dynamic target detection capability
LiDAR	Good at near range obstacle detection, high depth resolution and accuracy	Sensor noise and calibration errors, sensitive to environment and USV motion
Sonar	No visual restrictions, high depth resolution and accuracy	Limited detecting range in each scanning, impressionable to the noise from near surface
Visual sensor	High lateral and temporal resolution, simplicity and low weight	Low depth resolution and accuracy, challenge to real-time implementation, affected by light and weather conditions
Infrared sensor	Applicable in dark conditions, low power consumption	Indoor or evening use only, affected by interference and distance
IMU	Small size, low cost and power consumption	Affected by accumulated errors and magnetic environment
GPS	Small size, low cost and power consumption	Affected by loss or jamming of signals and magnetic environment

**Table 6 sensors-21-01518-t006:** Some examples of USV sensor application specifications.

**USV Application: General**		
**Functions**	**Sensors**	**Specifications**
Anti-collision	Radar	Range from 1.5 to 340 m Distance measurement accuracy 0.25 m FOV (horiz.) 100°/(vert.) 16° Weight 1290 g Size 230 × 160 mm
	LiDAR	Range 100 m FOV (horiz.) 30°/(vert.) 360° Weight 830 g Size Diam. 103 mm × height 72 mm
	Visual camera	Resolution 1920 × 1080; 3840 × 2160 Frame rate 60 fps; 30 fps Size 140 × 98 × 132 mm Weight 461 g
**USV Application: Requiring small low cost USV**		
**Functions**	**Sensors**	**Specifications**
Exploration	LiDAR	Range 5 cm to 40 m Accuracy +/− 2.5 cm at distances > 1 m Size 40 × 48 × 20 mm Weight 22 g
	Visual camera	Maximum resolution 3280 × 2464 Frame rate from 30 fps to 90 fps Size 25 × 20 × 9 mm
**USV Application: Archaeology**		
**Functions**	**Sensors**	**Specifications**
Bathymetric survey in shallow waters	Sonar	Depth range 0.30 m to 75.00 m Accuracy +/− 0.025 m (RMS) Operating temperature 0 to 45 °C Size 100 mm × 220 mm × 45 mm Weight 0.75 kg
Conditions and the presence of obstacles in real-time	Visual camera	Underwater depth up to ≈40 m Camcorder sensor res. 5 megapixels Video capture 1920 × 1080—30 fps; 1280 × 960—30 fps; 1280 × 720—60/30 fps; 848 × 480—60 fps Weight ≈ 0.74 kg
Obstacle detection	Ultrasonic	Detection range 2–450 cm Accuracy 0.2 cm
**USV Application: Detection and Tracking of oil spills**		
**Functions**	**Sensors**	**Specifications**
Locating oil spills on water	Thermal IR camera	FOV 18° × 13°; 30° × 23°; 32.4° ×25.6° Temperature resolution 0.05–0.1 °C
	Radar	Range 2–7 km FOV 360°

**Table 7 sensors-21-01518-t007:** Characteristics of UUV sensors [66].

Sensors Types	Capabilities	Challenge and Limitations
Inertial	Position, orientation and velocity information is carried out by collecting data from accelerometers and gyroscopes	Data processing and fusion of data from multiple sensors are required to correct for drift errors
Acoustic	Acoustic transponders are used to determine positioning relative to receivers or features (seafloor)	Fixed infrastructure can be required, constraints due to water environment, possible speed restrictions
Depth	The ambient pressure in the water column is measured to calculate depth	Limitations are minimal, measurement sensors will function at depths much greater than projected platforms are intended to go
Orientation	Platform heading is calculated from one or several sensors	Degraded performance during acceleration
Light and optical	Positioning is carried out using environmental features as a guide	Light attenuation in the water limits accuracy
Light and optical: light detection and ranging, laser line scanning	Laser mapping and imaging, video feed generation	The minimal optical wavelength propagation through water drives the requirement to be physically close to targets
Sonar: single beam, multibeam, sidescan, synthetic aperture	Target detection and identification, buried object detection, imaging	Sound propagation in water depends on the temperature and salinity, calibration is needed

**Table 8 sensors-21-01518-t008:** Some examples of UUV sensor application specifications.

**UUV Application: Long-Range Surveys**		
**Functions**	**Sensors**	**Specifications**
Docking guidance systems	Acoustic navigation sensor	Acquisition range in order of km (up to 2 km)
	Optical navigation sensor	Acquisition range 10~30 m Lateral and temporal accuracy 2~10 cm
	Electromagnetic navigation sensor	Acquisition range 25~30 m Accuracy ≈ 20 cm
**UUV Application: Collaborative missions, surveillance and intervention**		
**Functions**	**Sensors**	**Specifications**
Localization and navigation for collaborative work	Sonar	Accuracy from 5–10 cm to 10–120 cm Range From 5 m up to hundreds m from obstacles
	Acoustic range (LBL, SBL, USBL)	Accuracy from some cm up to tens of m Range Up to tens of m from array
	Optical light sensors	Accuracy Up to 20 cm for position and 10° for orientation Range 1–20 m from markers
	Optical cameras	Accuracy Up to 1 cm for position and 3° for orientation Range 1–20 m from markers
**UUV Application: Seafloor mapping**		
**Functions**	**Sensors**	**Specifications**
Seafloor topographical data, measure the size, shape, and height variation of underwater targets	Sonar (MBE)	Range 175–635 m Frequency 200–450 kHz Transmit beamwidth 0.4°–2.0°; Receive beamwidth 0.5°–2.0°System depth rating 6000 m
	Sonar (SSS)	Range 150–600 m Frequency 75–600 kHz Horizontal beams 0.2°–1.0° Depth rating 2000–6000 m

**Table 9 sensors-21-01518-t009:** Some technologies for MAV relative localization [78].

Sensor Technology	Advantages	Limitations
Vision (direct, passive, visual markers or colored balls)	Rich information; passive sensors	Dependent on light conditions; visual line of sight needed; computationally expensive
Vision (direct, with active markers, infrared or ultraviolet)	Can operate in visual cluttered environments; accurate; low dependence on light conditions	Visual line of sight needed; can result in high energy expense
Sound	Can operate in visual cluttered environments; omni-directional; passive sensors	Angular accuracy is limited due to limited baseline between microphones in the array; limited range
Infrared sensor array	Can operate in visual cluttered environments; accurate and computationally simple	Visual line of sight needed; heavy; can result in high energy expense

**Table 10 sensors-21-01518-t010:** Some examples of UAV swarm sensor application specifications.

**UAV Swarm Application: Aerial Surveillance, Remote Sensing, Aerial Inspections**		
**Functions**	**Sensors**	**Specifications**
Avoid collisions with other members of swarms and environmental obstacles	Infrared	Range 20 to 150 m Operating temperature −10 to +60 °C Size 29.5 × 13 × 21.6 mm
	Ultrasonic	Range 0 to 6.45 m Resolution 2.5 cm Size 2.2 × 2.0 × 1.6 cm
**UAV swarm Application: Indoor outdoor**		
**Functions**	**Sensors**	**Specifications**
Relative localization	Camera	Resolution 752 × 480 FOV (horiz.) 185° Maximum frame rate 93 Hz

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
