# Peer review of "Sensors and Measurements for Unmanned Systems: An Overview"

_sensors, 2021, doi:10.3390/s21041518_

Round 1

Reviewer 1 Report

Use an org chart to visually classify all the different unmanned system before describing them.

Modify Figure 1 to make it more real and more technical.  For example what is the application of putting a UGV on a really small island?  What is the meaning of a paper and leaves close to the quadcopter?  What is the meaning of having a school of fishes close to the UUV?

The number of applications in Table 1. should be numbered.

Table 2. is a small table and it could probably be combines with another table.

In Table 3. what are the approximate ranges of all the terms in quantitative terms, i.e. What is high accuracy? What is low accuracy? What is high cost? What is low cost?

Table 4. again mention include the advantages and limitations in quantitative terms.

Table 5. include values for the capabilities.

Table 6. include values for each technology.

Author Response

The authors thank the Editor and the Reviewers for the useful suggestions that improved the scientific content and readability of the manuscript.

In the following, the remarks of the Reviewers are reported in black, the corresponding authors’ answers and comments with details about the changes are reported in blue, while the added or modified text in the paper is reported in red to simplify the review.

Reviewer 1:

Use an org chart to visually classify all the different unmanned system before describing them.

Modify Figure 1 to make it more real and more technical.  For example what is the application of putting a UGV on a really small island?  What is the meaning of a paper and leaves close to the quadcopter?  What is the meaning of having a school of fishes close to the UUV?

The number of applications in Table 1. should be numbered.

The Figure 1 has been deleted and replaced with a different figure showing the classification of the different unmanned systems which are subject of the review paper, along with some examples of applications of them and environmental critical issues.

Table 2. is a small table and it could probably be combines with another table.

Table 2 has been extended including other proposed solutions for UAV indoor localization.

In Table 3. what are the approximate ranges of all the terms in quantitative terms, i.e. What is high accuracy? What is low accuracy? What is high cost? What is low cost?

Table 4. again mention include the advantages and limitations in quantitative terms.

Table 5. include values for the capabilities.

Table 6. include values for each technology.

As discussed in the added text to Section 3, it is very complex to identify absolute values for all the specifications and parameters describing all kind of sensor technologies that can be used in the different types of unmanned vehicles. Anyway, five new tables (Table 2, 4, 6,8 and 10) have been added where the main specifications of some typical sensors for different functions and applications of the 4 considered unmanned system platforms are reported.

Moreover, further details about the sensor range have been added to Table 3.

Reviewer 2 Report

The information in this article is very general. There are no sensor parameters such as measuring range, communication interface, dimensions, weight, which are important both for design and operation. In Fig. 2: Pictures of ToF Camera (Sensor Sharp) - the measuring principle of this device is different from ToF. Even the information about the sensors pictured is not included.

For the person who deals with the subject of measurements, the information provided is obvious, and there are no new measurement methods or data processing algorithms.

The article, although it contains very large literary references, does not add much in itself. Suggestion: rebuild the article to include specific information on important parameters of measuring devices, including names and example symbols of the device.

Author Response

The authors thank the Editor and the Reviewers for the useful suggestions that improved the scientific content and readability of the manuscript.

In the following, the remarks of the Reviewer are reported in black, the corresponding authors’ answers and comments with details about the changes are reported in blue, while the added or modified text in the paper is reported in red to simplify the review.

Reviewer 2:

The information in this article is very general. There are no sensor parameters such as measuring range, communication interface, dimensions, weight, which are important both for design and operation. In Fig. 2: Pictures of ToF Camera (Sensor Sharp) - the measuring principle of this device is different from ToF. Even the information about the sensors pictured is not included.

For the person who deals with the subject of measurements, the information provided is obvious, and there are no new measurement methods or data processing algorithms.

The article, although it contains very large literary references, does not add much in itself. Suggestion: rebuild the article to include specific information on important parameters of measuring devices, including names and example symbols of the device.

Figures 2, 3, 4 and 5 have been deleted as they did not give precise information on the sensor technologies used and the paper has been rebuilt as suggested to include sensor specifications, by adding five new tables (Table 2, 4, 6,8 and 10) where the main specifications of some typical sensors for different functions and applications of the 4 considered unmanned system platforms are reported.

Specific references to the papers describing the reported sensors have been added, too.

Round 2

Reviewer 1 Report

The authors addressed previous concerns.  The paper has more technical content and the authors also extended the number of references.  I think all these characteristics make a review paper on unmanned systems worth reading by professionals and researchers interested in learning about current state of the art on this topic.

Author Response

The authors thank the Reviewer for the appreciation of their work.

Reviewer 2 Report

Table 1: Stereo Vision  10 m or cm?

Table 6. New. line

Tab. 6, 8, 10: The last two columns are disproportionate to the rest. from these columns (Sensors Specifications) can be summarized in an additional table 

Author Response

The value has been corrected in the Table 1. All the Tables in the paper have been changed to make the included information easily and suitably readable.